# Application of Auxiliary Classifier Wasserstein Generative Adversarial Networks in Wireless Signal Classification of Illegal Unmanned Aerial Vehicles

**Caidan Zhao [1,\*], Caiyun Chen [1], Zeping He [1] and Zhiqiang Wu [2,3]**

[1] Department of Communication Engineering, Xiamen University, Xiamen 361005, China; chency@stu.xmu.edu.cn (C.C.); 23320181154304@stu.xmu.edu.cn (Z.H.)

[2] Department of Communication Engineering, Tibet University, Lhasa 850000, China

[3] Department of Electrical Engineering, Wright State University, Dayton, OH 45435, USA; zhiqiang.wu@wright.edu

\* Correspondence: zcd@xmu.edu.cn; Tel.: +86-592-258-0078

**Abstract:** Recently, many studies have reported on image synthesis based on Generative Adversarial Networks (GAN). However, the use of GAN does not provide much attention on the signal classification problem. In the context of using wireless signals to classify illegal Unmanned Aerial Vehicles (UAVs), this paper explores the feasibility of using GAN to improve the training datasets and obtain a better classification model, thereby improving the accuracy of classification. First, we use the generative model of GAN to generate a large datasets, which does not need manual annotation. At the same time, the discriminative model of GAN is improved to classify the types of signals based on the loss function of the discriminative model. Finally, this model can be used to the outdoor environment and obtain a real-time illegal UAVs signal classification system. Our experiments confirmed that the improvements on the Auxiliary Classifier Generative Adversarial Networks (AC-GANs) by limited datasets achieve excellent results. The recognition rate can reach more than 95% in the indoor environment, and this method is also applicable in the outdoor environment. Moreover, based on the theory of Wasserstein GANs (WGAN) and AC-GANs, a more robust Auxiliary Classifier Wasserstein GANs (AC-WGANs) model is obtained, which is suitable for multi-class UAVs. Through the combination of AC-WGANs and Universal Software Radio Peripheral (USRP) B210 software defined radio (SDR) platform, a real-time UAVs signal classification system is also implemented.

**Keywords:** GAN; AC-WGANs; wireless signals; classify model; USRP

## 1. Introduction

Since the theory of Generative Adversarial Networks (GAN) [1] was proposed in 2014, GAN has been widely used in various fields [2]. In signal synthesis, Radford et al. [3] used the DCGAN to generate various image datasets, and Yang et al. [4] used GAN for music generation. In signal translation, Cycle-GAN is used to perform image-to-image translation [5,6]. GAN is also used for text-to-image generation [7], texture synthesis, style transfer, video stylization [8], pixel-level domain transfer [9], image inpainting [10] and so on. Since GAN has a generative model *G* and a discriminative model *D*, we explore in this paper whether the discriminative model of GAN can effectively be used for signal classification problem as well. In signal classification, a typical challenge is that the labeled data are hard to collect and therefore limited. As a direct consequence, the training result is significantly affected. To solve this problem, we explore using the GAN model to improve the training datasets and recognition model to obtain better classification results in the wireless signal recognition of Unmanned Aerial Vehicles (UAVs).

In recent years, with the rapid development of the UAV technology, civilian and consumer-grade UAVs have been put into use on a large scale, including disaster rescue [11], delivery service [12] and so on. However, reports of unmanned aircraft (UAS) sightings from pilots, citizens, and law enforcement also have increased dramatically over the past two years, the FAA now receives more than 100 such reports each month [13]. UAVs have crashed into the White House, a nuclear power station [14], an airport [15] and many other places, buildings and infrastructure have been destroyed and many innocent people have been hurt [16]. However, due to the extensive application of UAVs [17], the scene of multiple UAVs flying is inevitable [18]. As shown in Figure 1, it has become a necessity to distinguish illegal UAVs from legal ones when they are flying together.

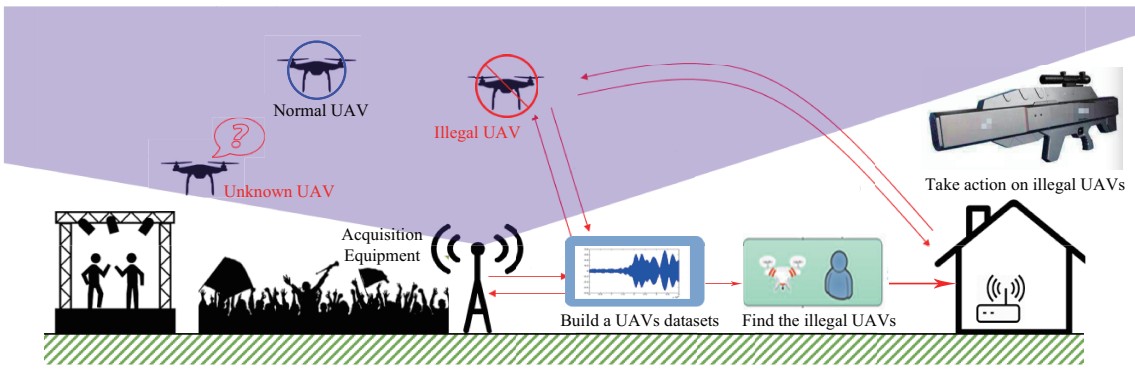

**Figure 1.** The application UAVs classification systems in the scene of multiple UAVs.

UAVs in operation are conventionally detected by radar, visual detection or acoustic sensors, as shown in Table 1. Park and Park [19] discussed the detection of UAVs with Frequency-Modulated Continuous-Wave (FMCW) Radar system, and the UAV signal was detected up to more than 500 m of distance in real-time with the error of less than 0.1%. However, the cost of radar detection is relatively high, and it is difficult to detect at a lower height. Nguyen et al. [20] introduced a system for detecting the presence of UAVs by identifying unique signatures of a UAV's body vibration and body shifting in the WiFi signal. The recognition rate of this method approximately reached 80% at a distance of 600 m. Jeon et al. [21] presented a method capable of detecting the presence of commercial hobby UAVs as a binary classification problem based on sound event detection. This method used Recurrent Neural Network (RNN), which showed the best detection performance with an F-Score of 0.8009 with 240 ms of input audio with a short processing time, indicating the applicability to real-time detection systems. However, this system is easy to be deceived and cannot identify some types of UAVs. Lim et al. [22] proposed a system that classifies payload carrying and non-payload carrying DJI Phantom II UAVs by presenting sound spectrum data to a simple Convolutional Neural Networks (CNN). These networks provide about 99.92% recognition rate for this problem without the need to violate minimal cost constraint, but video detection is easily affected by weather and lighting. Unlu et al. [23] used a two-dimensional scale, rotation and translation invariant Generic Fourier Descriptor (GFD) features and classified targets as a drone or a bird by a neural network. This system can achieve up to 85.3% overall correct classification rate. However, this method is unable to deal with the micro-Doppler effect. Richardson [24] provided a detection method based on Medium Access Control (MAC) address given the fact that many low-end commercial UAVs have identifiable service set identifier (SSIDs) and MAC address broadcasting. However, this MAC address based detection method is vulnerable to interference, and wireless protocols must be known beforehand.

**Table 1.** Summary of existing UAV detection technologies.

| Detection Technology | Advantages | Drawbacks |
| --- | --- | --- |
| Radar detection | Well-Suited for long-distance. No need cooperation from the target. | Expensive. Low/no waves reflect from non-reflective materials. Hardly distinguishable from birds or bats. Flying at an altitude of fewer than 100 feet would be difficult to detect. |
| Acoustic detection | Low cost.Passive. Easy to be combined with other technology. | Easy to be deceived.Cannot detect fixed wing UAV. Detection distance less than 500 m. |
| Visual detection | Flexibility. Inexpensive.Detect either large aircraft or small objects. | Limited by light and weather. Need to create recognition database. |
| MAC address | Relatively inexpensive. Effective detection and accurate tracking. | Suffer from interference. Must have some knowledge of emitter parameter and protocol. |
| Ray Tracing Simulations | Convenient to simulate different scenarios. | Unable to deal with micro-Doppler effect. |

Zhang et al. [25] proposed a detection algorithm based on an Artificial Neural Network (ANN) where the recognition rate is greater than 82% within a distance of 3 km. Fu et al. [26] presented an SDR-based, portable universal software radio peripheral (USRP) system for detection in two scenarios. For the scenario in which a UAV communicates with the ground controller, the cyclostationarity signature of the drone signal and pseudo-Doppler principle are employed. For the scenario in which a UAV is not sending any signal, a micro-Doppler signature generated by the radio frequency signal is exploited for detection and identification. Bisio et al. [27] proposed a WiFi-based approach that aimed at detecting nearby aerial or terrestrial devices by performing statistical fingerprint analysis on wireless traffic.

However, most of the above-mentioned methods are not very robust and cannot detect illegal UAVs. Due to the small size, low flying height and slow speed of most UAVs, most of existing detection systems do not perform well. Small UAVs are not easy to distinguish from birds and can be hidden by urban buildings, therefore the detection technology has a high false positive rate. In this paper, an SDR-based wireless signal detection technology is used to collect the wireless communication signals of the UAVs, and is then combined with an Auxiliary Classifier Wasserstein Generative Adversarial Networks (AC-WGANs) classification model to realize low-cost, high-precision UAV detection and recognition. Due to the crystal oscillation of the device, component jointing, and the loss of electronic device, the radio frequency (RF) signals of the device are slightly different, and thus can be distinguished. The technology based on wireless signals detection has been used to classify a number of Internet-of-Things (IoT) devices (same model and same RF front-end), which proves the feasibility of this method to detect the UAVs [28]. The technology based on wireless signal detection of UAVs has two main challenges. On the one hand, a variety of wireless signals exist in the real environment. As a direct result, it is very difficult to accurately collect the UAVs signals, label the signals, and establish a reliable database. On the other hand, the classification models such as k-Nearest Neighbor (KNN), Convolutional Neural Networks (CNN), and Support Vector Machine (SVM) need to select the appropriate feature signals for classification through feature extraction. In the process of feature selection, important feature information is often lost, and the recognition rate of the signal is seriously affected. To solve these two challenges, we use the combination of oscilloscope and USRP to acquire the signals of UAVs in the indoor environment and the outdoor environment to establish the database. Next, we propose an automatic classification model of the UAV based on AC-WGANs. Without special feature extraction, only a modified principal Component Analysis (PCA) reduction operation is needed to realize the automatic classification of the UAVs signals.

The contributions of this paper are as follows:

- We propose a novel method of using AC-WGANs model to classify the wireless signal to identify illegal UAVs. Specifically, we improve the generative model *G* to enhance the training datasets and use the loss function of the model to classify the wireless signals.
- We establish an illegal UAVs classification system based on the wireless signal. The system is divided into three parts: collecting the wireless signal, pre-processing, and classification. Combined with AC-WGANs model and USRP, a real-time classification system is implemented.
- We compare and analyze the UAVs signals in the indoor environment and the outdoor environment. The recognition rate can reach more than 95% in the indoor environment, and this system is also suitable for outdoor environments.

The rest of the paper is organized as follows. In Section 2, we introduce the related work of the wireless signal classification system and the basis of the AC-WGANs model. In Section 3, we provide a network overview of the AC-WGANs model. In Section 4, we analyze the difference of the indoor environment and outdoor environment and propose a real-time classification system. Finally, conclusions and future work are drawn in Section 5.

## 2. Related Work

The related work includes two parts: the characteristics of the wireless communication signal of UAVs and the basis of the AC-WGANs classify model.

### 2.1. Wireless Signals of UAVs

Most UAVs use wireless signals for communication (e.g., Phantom, Parrot, Hubsan, Xiro and so on [29,30]). Current 802.11 standards specify frame types for use in transmission of data as well as management and control of wireless links, and frames are divided into very specific and standardized sections. Each frame consists of a MAC header, payload, and frame check sequence (FCS) [31]. The antenna orientation in relation to the frame of the UAV (single and multiple antennas) influences the link quality and range of communication significantly. Using a three antenna extension to IEEE 802.11 devices, air-to-ground links can support high throughput at distances up to 500 m. Therefore, we can use the wireless signals to detect UAVs, despite the size, line of sight, protocol standardization and forensic tool support (hardware and software) of the UAVs. Almost all UAVs use the standard Institute of Electrical and Electronics Engineers (IEEE) 802.11a/b/g/n/w protocol to pair with controller devices, and the working frequency is between 2.4 and 2.5 GHz. All standards have the same preamble that signifies the start-point of a signal, and the center frequency of the transmission. At the 2.4 GHz band, WiFi is allocated throughput 11 wireless channels ranging from 1 to 11. However, only three of these channels (1, 6, and 11) are used, as each channel is only 5 MHz wide. Since transmissions on the 2.4 GHz band are 20 MHz wide, the utilized channels are spaced to mitigate co-channel interference [32]. Table 2 provides a summary of the frequency, modulation, and technology systems used by most of the UAVs, including DJI phantom, Parrot, Spektrun, JR, Futaba and so on.

**Table 2.** Popular technology systems used by UAVs.

|  | Frequency | Modulation | Technology |
|---|---|---|---|
| Control | 2.4 GHz/5.8 GHz | FHSS/DSSS/OFDM | FASST/Lightbridge/DSMX /DMSS/AFHSS/HOTT /ZigBee/WiFi |
| Telemetry | 868 MHz/433 MHz/ 2.4 GHz/4 GHz | Divers/DSSS/OFDM | ZigBee |
| Video | 2.4 GHz/5.8 GHz | OFDM/FM | Lightbridge/WiFi |

In this paper, we use the USRP software defined radios, oscilloscope, and antennas to collect UAVs wireless signals. Different UAVs have different wireless signals. For example, the center frequency

of Phantom2 operation is 2.447 GHz, and the center frequency of Phantom3 operation is 2.457 GHz, as shown in Figure 2. Thus, when we collect the UAVs signal, the center frequency and bandwidth of the acquisition system should be accurately set in the acquisition. At the same time, by comparing the wireless signals of different UAVs, it is evident that the signals of different UAVs have different features.

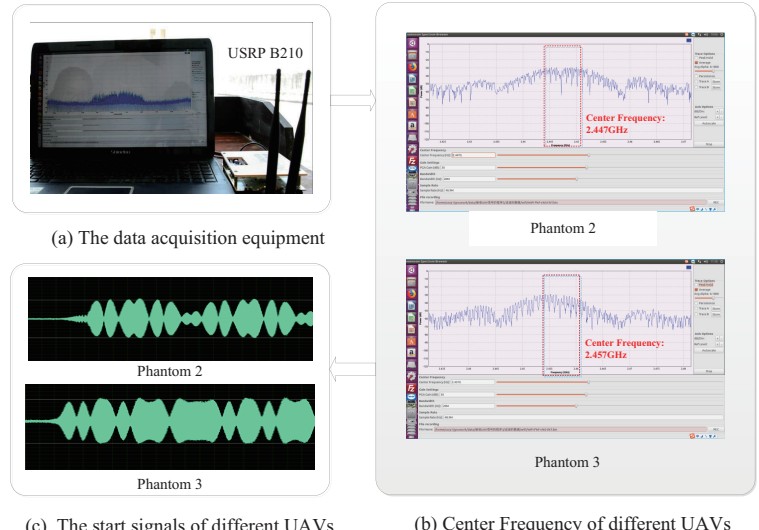

(a) The data acquisition equipment

(c) The start signals of different UAVs

(b) Center Frequency of different UAVs

**Figure 2.** (**a**) The data acquisition equipment USRP B210. (**b**) The center frequency of different UAVs including Phantom2: 2.447 GHz, and Phantom3: 2.457 GHz. (**c**) The wireless communication signals of different UAVs.

## 2.2. The Evolution of GAN Model

The GAN model includes a generative model *G* and a discriminative model *D*, which uses an adversarial process to propose a new framework for estimating generative models [1]. We can improve the *D* model to classify the wireless signals collected from the UAVs. Since 2014, GAN has appeared in various forms, as shown in Figure 3. In 2015, Radford et al. [3] found a Deep Convolutional GAN (DCGAN) model, which is more stable in training and produces higher quality samples. The DCGAN is competitive with a probabilistic generative data augmentation technique utilizing learned per class transformations [33] while being more general as it directly models the data instead of transformations of the data. However, the DCGAN model is unsuitable for classification. Information Maximizing GAN (InfoGAN) proposed by Chen et al. [34] uses the information-theoretic extension to learn disentangled representations in a completely unsupervised manner. In contrast to previous approaches that require supervision, InfoGAN is completely unsupervised and learns interpretable and disentangled representations on challenging datasets. In addition, InfoGAN adds only negligible computation cost on top of GAN and is easy to train. However, the core idea is more suitable for learning hierarchical latent representations, improving semi-supervised learning, and using InfoGAN as a high-dimensional data discovery tool. In 2016, Mao et al. [35] found that regular GANs hypothesize the discriminator as a classifier with the sigmoid cross entropy loss function. However, this loss function may lead to the vanishing gradients problem during the learning process. The Least Squares Generative Adversarial Networks (LSGANs) adopt the least squares loss function for the discriminator. Minimizing the objective function of LSGAN yields minimizing the Pearson $\chi^2$ divergence. The LSGANs are able to generate higher quality images than regular GANs and are more stable during the learning process. However, there is little research of the LSGANs in complex datasets.

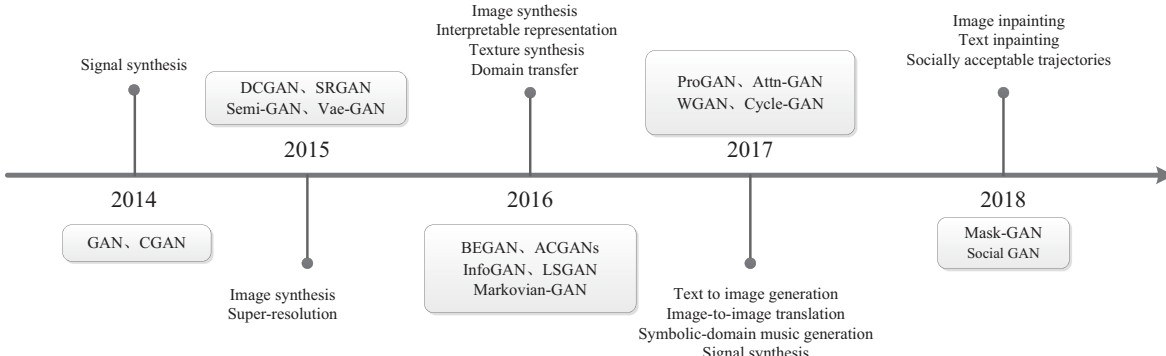

**Figure 3.** The evolution of GAN model.

Wasserstein GAN (WGAN) [36] can improve the stability of learning, get rid of problems such as mode collapse, and provide meaningful learning curves useful for debugging and hyperparameter searches. Furthermore, the corresponding optimization problem is sound and provides extensive theoretical work, highlighting the deep connections to different distances between distributions [37]. The Auxiliary Classifier GANs (AC-GANs) proposed by Odena et al. [38] can generate multiple samples, which expand on previous work for image quality assessment to provide two new analyses for assessing the discriminability and diversity of samples from class-conditional image synthesis models. Most of the GAN models are used for image sample generation rather than for classification and recognition detection. In our paper, we improve the discriminative *D* model of the AC-GANs model, which can be adapted to the multi-class identification and detection of wireless signals, and enhance the AC-GANs model according to the WGAN model. Therefore, a more stable AC-WGANs model can be obtained to effectively enhance the recognition rate. The framework of the AC-WGANs model is shown in Figure 4.

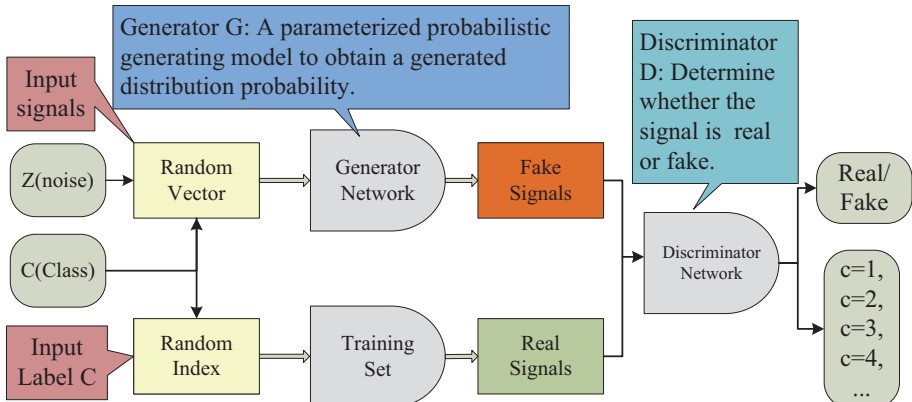

**Figure 4.** The framework of AC-WGANs model.

## 3. Network Overview

The AC-GANs model is a method commonly used for image synthesis [38] and the model can be improved as a multi-classification recognition model. In the AC-GANs, every generated sample has a corresponding class label, $c \sim p_c$, in addition to the noise, $z$. $G$ uses both to generate signals $X_{fake} = G(c, z)$ and real signals $X_{real} \sim \tilde{x}(i)$. The discriminator gives both a probability distribution over sources and a probability distribution over the class labels, $P(S|X), P(C|X) = D(X)$. The objective function has two parts: the likelihood of the correct source $L_S$, and the likelihood of the correct class $L_C$.

$$L_S = E[\log P(S = real | X_{real})] + E[\log P(S = fake | X_{fake})] \tag{1}$$

$$L_C = E[\log P(C = c|X_{real})] + E[\log P(C = c|X_{fake})] \tag{2}$$

$D$ is trained to maximize $L_S + L_C$ while $G$ is trained to maximize $L_C - L_S$ [38].

We input the training data to train the $D$ and $G$ models. After that, we use the parameter of the $D$ model to create a test model $D'$ to classify the test signals, and obtain the likelihood of the correct class $L'_C$.

$$L'_C(m)^M_{m=1} = E[\log P(C = c'(m)|X'_{real})]^M_{m=1} \tag{3}$$

where $X'_{real} \sim \tilde{x}(m)$ is the testing data, $M$ is the type of wireless signals, and $c'(m)$ is the label of each type of wireless signals. Next, we compare the $L'_C$ to get the most likely label as the test label of the testing data and obtain the recognition rate.

WGAN [36] and improved WGAN [39] use Kullback–Leibler (KL) instead of Jensen–Shannon (JS) divergence. WGAN can not only improve the stability of learning and reduce problems such as mode collapse but also provide meaningful learning curves that are useful for debugging and hyperparameter searches.

$$\min_G \max_{D \in \mathbb{D}} \underset{x \sim Pr}{E}[D(X_{real})] - \underset{x \sim Pg}{E}[D(X_{fake})] \tag{4}$$

WGAN achieves the stable state of the model by finding the minimum loss function of the generated model $G$ and discriminating the maximum loss function of the model $G$. Combined with the AC-GANs multi-classification model, the specific practices include the following three points [36,39]:

1.  In the feed-forward neural network, it is unnecessary to use smooth Lipschitz functions such as the *sigmoid*.

$$g_w = \nabla_w[\frac{1}{m}\sum_{n=1}^{m} f_w(c, g_\theta(X_{real})) - \frac{1}{m}\sum_{n=1}^{m} f_w(c, g_\theta(X_{fake}))] \tag{5}$$

$$g_\theta = -\nabla_\theta \frac{1}{m}\sum_{n=1}^{m} f_\theta(c, g_\theta(X_{fake})) \tag{6}$$

2.  *RMSProp* is chosen instead of *Adam*, which is known to perform well even on very non-stationary problems ($\alpha$ is the learning rate).

$$w = w + \alpha \cdot RMSProp(\theta, g_w) \tag{7}$$

$$\theta = \theta - \alpha \cdot RMSProp(\theta, g_\theta) \tag{8}$$

3.  The *log* trick can be ignored when training the GAN model with a convolutional architecture.

$$L_S = E[P(S = real|X_{real})] + E[P(S = fake|X_{fake})] \tag{9}$$

$$L_C = E[P(C = c|X_{real})] + E[P(C = c|X_{fake})] \tag{10}$$

$$L'_C(m)^M_{m=1} = E[P(C = c'(m)|X'_{real})]^M_{m=1} \tag{11}$$

## 4. Experiments

As shown in Figure 5, the classification system can be used to protect the important areas by separating the illegal ones from all UAVs. The system based on wireless signals requires equipment to collect the UAV signal, so the detection range is also limited by the performance of the equipment device, including the range of the receivable signal, signal quality and so on. Therefore, in the case of limited acquisition equipment, we can set different application scenarios to adapt to different application environments. In some place, multi-point layout, fixed-point deployment, and multiple wireless signal acquisition devices can be adopted. The wireless signals collected by the signal acquisition device are then transmitted to the centralized processing system for processing. At last,

the processed information is transmitted through the UAVs cloud platform. In some areas, the data are relatively large, and the control scene can be used to collect wireless signals from illegal UAVs and controllers by means of irregular wireless patrols. In this way, equipment costs can be reduced. The classification system can be realized in real-time, including training the classification model of the signal centralized processing system and real-time test in the indoor environment and the outdoor environment.

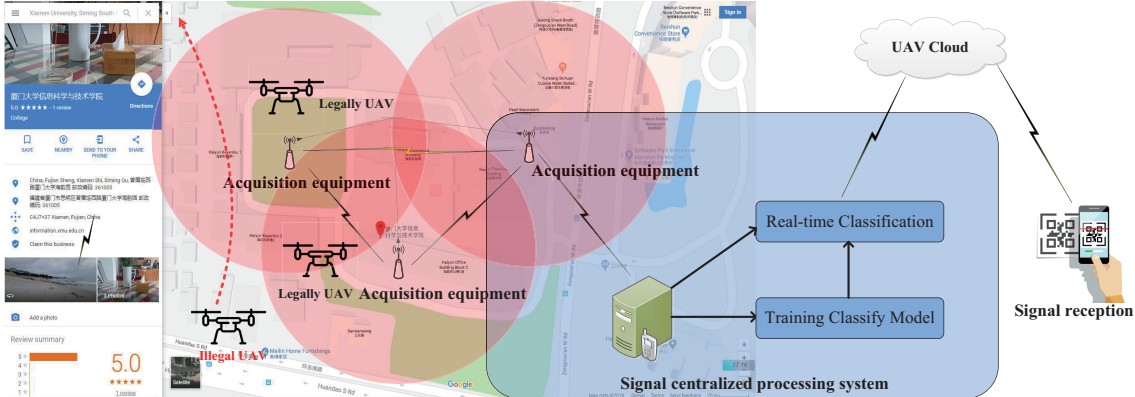

**Figure 5.** The application of the UAV classification system.

### 4.1. Indoor Environment

In the indoor environment, the situation of wireless signals is very complicated, especially because the wireless signals in the 2.4–2.5 GHz frequency range also include IEEE 802.11b and IEEE 802.11n WiFi signals. Thus, we first analyzed the wireless signals ranging from 2.4 to 2.5 GHz. We used an Agilent (DSO9404A) oscilloscope to collect the wireless signal of different devices. The oscilloscope has a sampling bandwidth of 1 GHz and a sampling rate of 20 GSa/s, which meets the sampling requirements [40]. It can be found that, even if the same WiFi protocol is used, different mobile phone models and different devices, such as routers, have different features. Figure 6a shows the signal acquisition process of the oscilloscope. The collected signal was detected to obtain the processed start-point signal. Figure 6b shows the signal waveforms of different devices. It can be found that different devices have different waveforms generated by different WiFi protocols.

Most of the UAV and WiFi signals are in the frequency range of 2.4–2.5 GHz, and the preamble of the wireless signal is identifiable and can be used for classification detection of the UAVs [41]. We selected five types of UAV signals from different manufacturers for detection and identification: Phantom, Hubsan, Mi, WiFi, and Xiro. Bandpass filtering, start-point detection [42] and dimensionality reduction were performed. After that, the model was trained and identified in the AC-WGANs model. The AC-WGANs algorithm flow includes four steps: signal acquisition, signal preprocessing, model training, and classification (Figure 7). The signal acquisition process of the UAV will cause some low-frequency noise due to factors such as environment and equipment stability. Therefore, the noise was filtered by setting a bandpass filter in the 2.4 GHz band, and then the start-point detection was performed. Finally, extracting the envelope data and reducing the dimensionality of the wireless signals were performed..

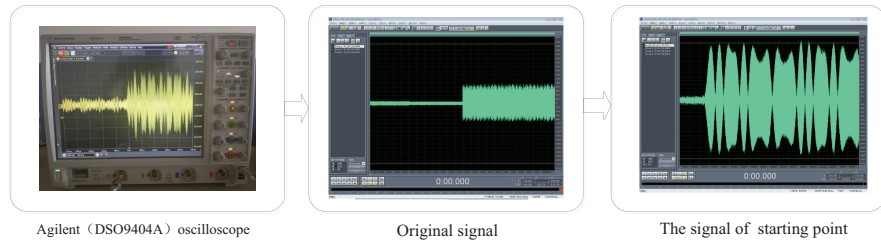

Agilent（DSO9404A）oscilloscope     Original signal     The signal of starting point

(a) The flowchart of signal collection

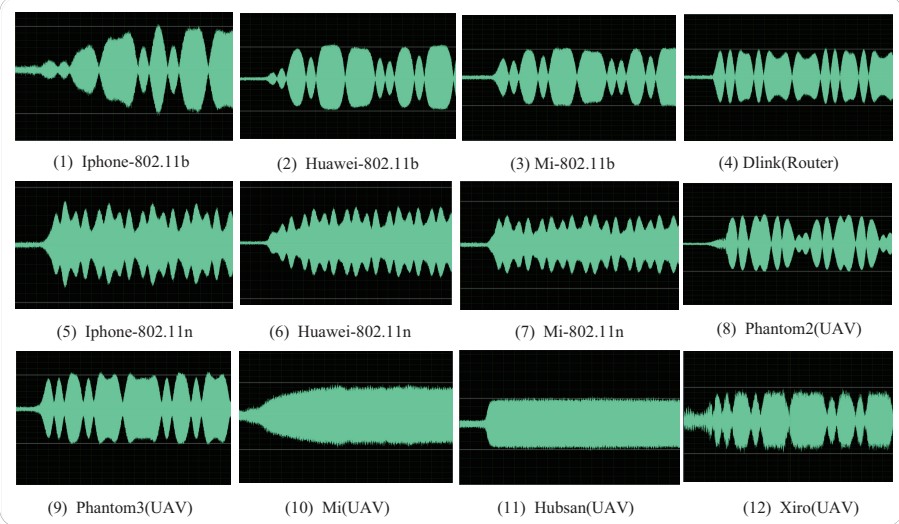

(1) Iphone-802.11b     (2) Huawei-802.11b     (3) Mi-802.11b     (4) Dlink(Router)

(5) Iphone-802.11n     (6) Huawei-802.11n     (7) Mi-802.11n     (8) Phantom2(UAV)

(9) Phantom3(UAV)     (10) Mi(UAV)     (11) Hubsan(UAV)     (12) Xiro(UAV)

(b) The wireless signals of different devices

**Figure 6.** Detecting the application scene of the UAVs.

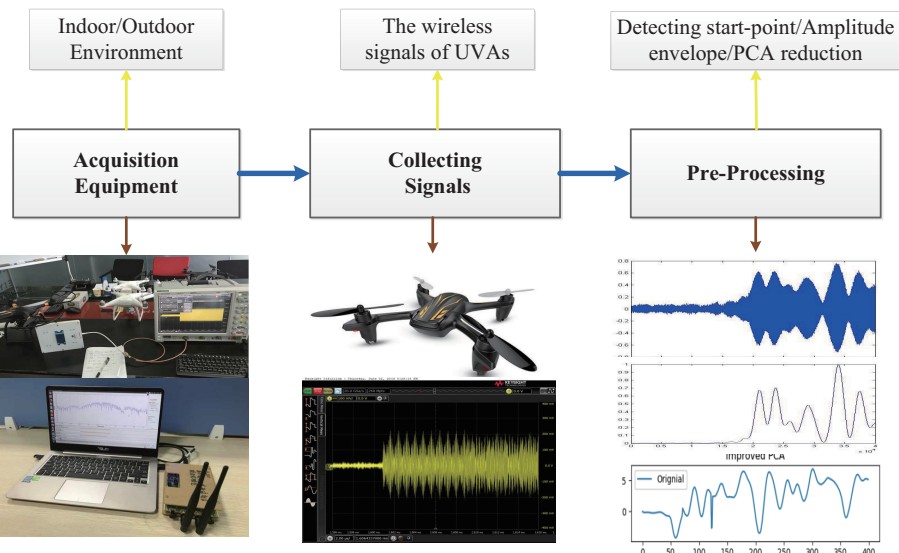

**Figure 7.** The pre-processing of the wireless signals.

After the pre-processing of the wireless signals, the AC-WGANs classify model was used for classification. The UAVs signal and WiFi signal was input to this model and the probability of each label was compared to determine the type of the inputting signal. To analyze the influence of noise on model recognition rate, artificial noises with a signal-to-noise ratio (SNRs) of 5 dB to 35 dB were added in the wireless signals to compare the effects of different noises on model recognition rate. Figure 8 shows the classification rate of wireless signals in different SNRs. In the case of low SNRs,

especially at 5 dB, the recognition rate was significantly improved, reaching more than 95%, and the signal may be classified as an unknown signal if the signal isn't in the datasets of the classify model, as shown in Table 3. Figure 8a shows the classification of AC-WGANs based on PCA and Restricted Boltzmann Machine (RBMs) [43]. In addition to the PCA, the RBMs was used in the dimensionality reduction, and also had good performance. However, considering the time complexity and space complexity, the PCA is more suitable. Figure 8b shows the comparison of the recognition rates of various algorithms, including Deep Belief Nets (DBN) [44], KNN, SVM, and GANs. Considering the stability and recognition rate, it can be seen that the improved AC-WGANs is better.

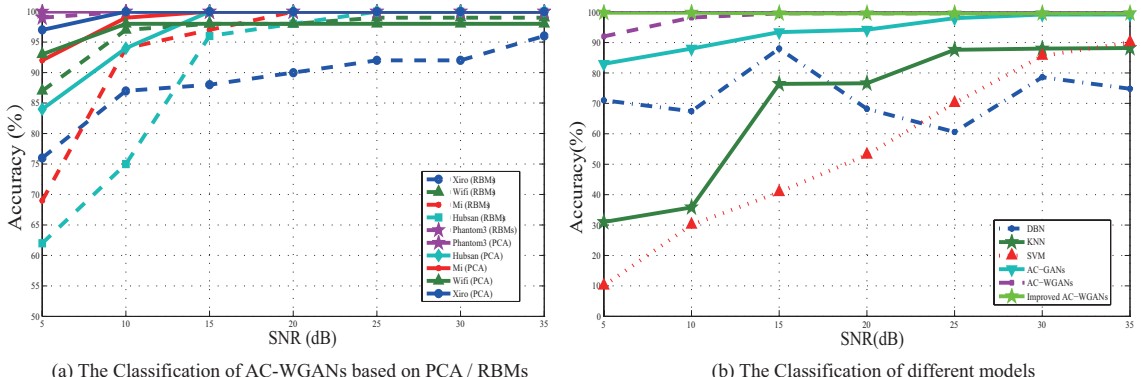

(a) The Classification of AC-WGANs based on PCA / RBMs      (b) The Classification of different models

**Figure 8.** The classification of indoor environment.

**Table 3.** Classification of different signals at 5 dB: confusion matrix.

|          | Phantom3 | Hubsan | Mi    | Wifi   | Xiro  |
|----------|----------|--------|-------|--------|-------|
| Phantom3 | 86.0%    | 0.0%   | 0.0%  | 0.0%   | 0.0%  |
| Hubsan   | 0.5%     | 99.5%  | 0.0%  | 0.0%   | 0.0%  |
| Mi       | 0.0%     | 0.0%   | 99.5% | 0.0%   | 0.0%  |
| Wifi     | 0.0%     | 0.0%   | 0.0%  | 100.0% | 0.0%  |
| Xiro     | 1.5%     | 0.0%   | 0.0%  | 0.0%   | 99.5% |
| Unknown  | 12.0%    | 0.5%   | 0.5%  | 0.0%   | 0.5%  |

## 4.2. Outdoor Environment

The outdoor environment includes different types of wireless signals in each frequency band. The oscilloscope can collect signals in the whole frequency range, and the long-distance UAVs signal strength is weak, so the wireless signal acquisition combined with the oscilloscope and the antenna is no longer sufficient. According to the frequency characteristics of the UAVs signal, the USRP B210 is suitable for collecting signals, which has a radio frequency range from 70 MHz to 6 GHz, supports a maximum real-time bandwidth of 56 MHz and has a reference sampling rate of 61.44 MS/s [45]. After collecting long strings of wireless signals through USRP, the signals were processed through IQ combination and start-point detection, and the useful signals were intercepted. The different kinds of signals in the outdoor environment are shown in Figure 9. Figure 9a is the signal acquisition equipment: USRP B210. Figure 9b,c shows the original and the amplitude envelope of UAVs wireless signals. Figure 9d shows four types of different wireless signals, including UAVs signal, unknown signal, and unexpected signal.

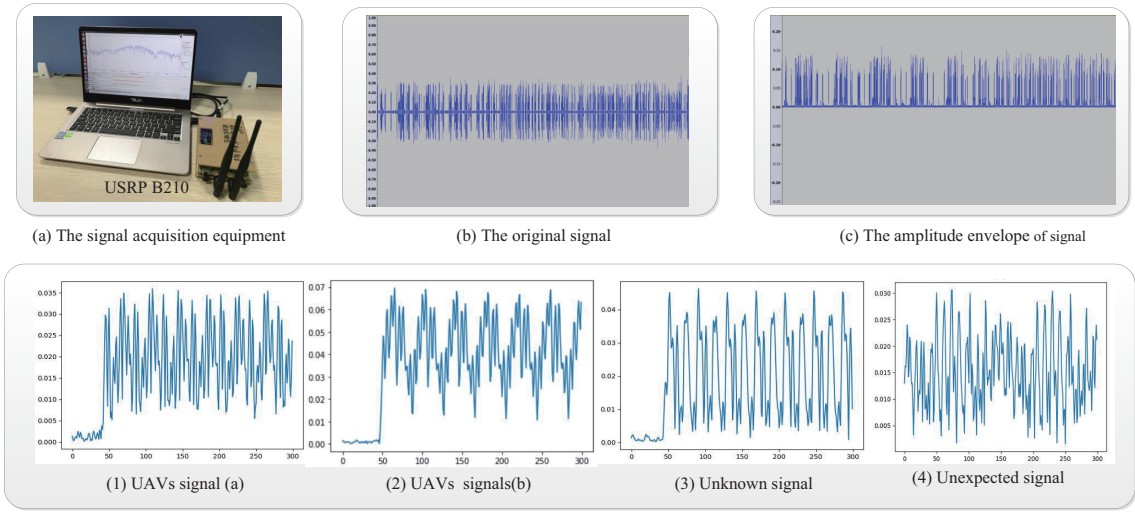

(a) The signal acquisition equipment    (b) The original signal    (c) The amplitude envelope of signal

(1) UAVs signal (a)    (2) UAVs  signals(b)    (3) Unknown signal    (4) Unexpected signal

(d) Different kinds of wireless signals

**Figure 9.** Different kinds of signals in the outdoor environment.

After setting the parameters of the USRP acquisition signal, the wireless signals of the flying UAVs could be collected. After pre-processing, various signals were separated, including various UAVs signals, unknown signals, and other unexpected signals. To train the AC-WGANs classification model, unsupervised learning such as K-means could be used to separate the signals, as shown in Figure 10. In the outdoor environment, the UAV signals of distances in the range of 10 m to 400 m were collected. After training the classification model, the USRP B210 collected the wireless signal in real time and performed pre-processing. The processed signal was input into the discriminative model *D* of AC-WGANs to classify and detect the wireless signals. The recognition results are shown in Figure 11. Figure 11a is the real-time testing in the outdoor environment, and Figure 11b is the classification of different distances.

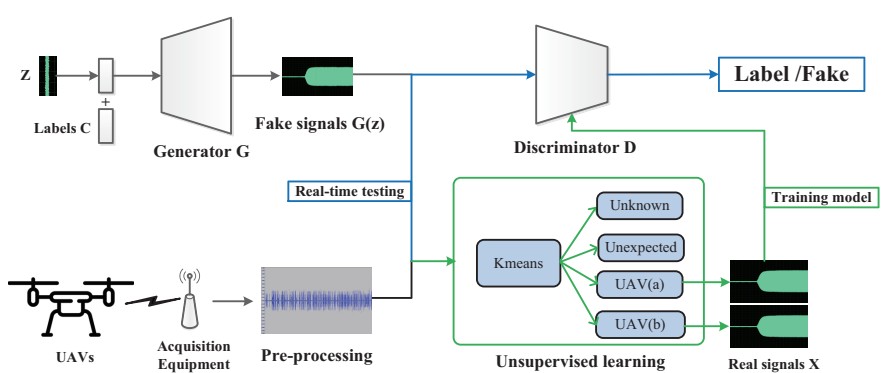

**Figure 10.** The real-time classification system of the UAV signal.

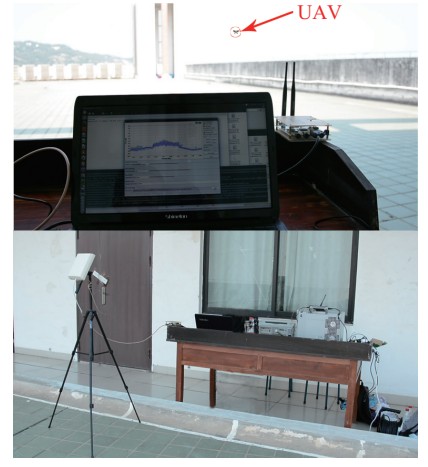

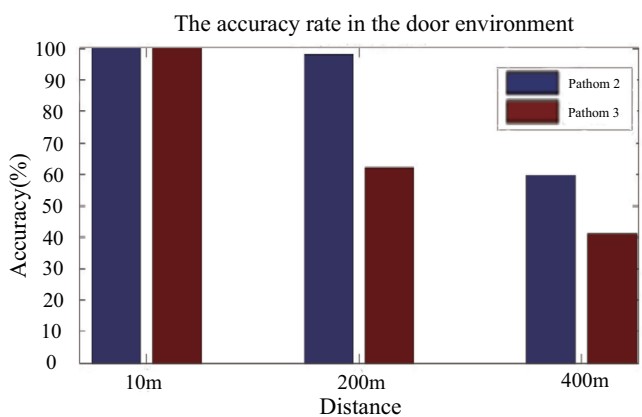

(a) The real-time testing in the outdoor environment      (b) Classification of different distance in the outdoor environment

**Figure 11.** The classification of different distance in the outdoor environment.

## 5. Conclusions

The wireless signals of UAVs are identifiable and can be used for detection of illegal UAVs. The AC-WGANs model has a high recognition rate and can be applied to the real-time classify system of the UAV signals in a long distance from 10 m to 400 m. The real-time classification system of the UAVs implemented in this paper can be used in the scene of multiple UAVs to separate the illegal ones. When combined with multiple pieces of acquisition equipment, this system can protect some important areas such as the airports and nuclear power stations. It should be very interesting to extend this framework to other domains such as video and audio classification. In the future work, the scenario of large datasets and the classification system of large datasets would be considered, and the case of similar devices (same model or same RF front-end), multiple UAVs would be analyzed, too. In this classification system, the influence of different models, SNRs and pre-processing methods would be detailed in the future work.

**Author Contributions:** C.Z. presented the idea and designed the proposed algorithm; C.C. built the simulation model and performed the simulation; Z.H. wrote the first draft of the manuscript; and Z.W. revised the manuscript.

**Funding:** This research was founded by the National Natural Science Foundation of China under Grant No. 91638204, the NSF Award 1748494 and Ohio Federal Research Network.

**Acknowledgments:** The authors would like to express their acknowledgement for the support from the National Natural Science Foundation of China under Grant No. 91638204.

**Conflicts of Interest:** The authors declare no conflict of interest.

## Abbreviations

The following abbreviations are used in this manuscript:

| | |
|---|---|
| AFHSS | Adaptive Frequency Hopping Spread Spectrum |
| Attn-GAN | Attentional Generative Adversarial Networks |
| BEGAN | Boundary Equilibrium Generative Adversarial Networks |
| CGAN | Conditional Generative Adversarial Nets |
| Cycle-GAN | Cycle-Consistent Adversarial Networks |
| DSMX | Digital Signal Multiplex Equipment |
| DSSS | Direct Sequence Spread Spectrum |
| FASST | Flexible Audio Source Separation Toolbox |

| FHSS | Frequency Hopping Spread Spectrum |
| --- | --- |
| FM | Frequency Modulation |
| Markovian-GAN | Markovian Generative Adversarial Networks |
| MASK-GAN | Masked Generative Adversarial Networks |
| OFDM | Orthogonal Frequency Division Multiplexing |
| ProGAN | Progressive Generative Adversarial Networks |
| Semi-GAN | Semi-supervised Learning Generative Adversarial Networks |
| SRGAN | Super-Resolution Generative Adversarial Networks |
| Vae-GAN | Variational Autoencoder Generative Adversarial Networks |

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
