# Peer review of "Application of Auxiliary Classifier Wasserstein Generative Adversarial Networks in Wireless Signal Classification of Illegal Unmanned Aerial Vehicles"

_applsci, doi:10.3390/app8122664_

Round 1

Reviewer 1 Report

In relation to the document format and spelling it is very well written there is nothing very important to point out and the presentation quality is very good. The paper presents the use of the generative adversarial networks for signal classification of know and unknow RF signals. The algorithm results could be considered good nevertheless, in my opinion, the results will be worst if a large data set is considered. Another important point that was not address in the document was the case of similar devices (same model or same RF front-end) it seems, from what I could understand, that this scenario is impossible to detect the present of both… Please clarify this scenario in the paper. Another important point that should be address by authors is the near-far impact in the algorithm accuracy, e.g. if the legal UAV is near to the RF receiver (detection system) the algorithm will detect an illegal UAV far from the receiver? Please clarify this situation.

Author Response

Dear Editors and Reviewer :

Thank you for your letter and for the reviewers’ comments concerning our manuscript entitled “Application of Auxiliary Classifier Wasserstein Generative Adversarial Networks in Wireless Signal Classification of Illegal Unmanned Aerial Vehicles” (ID: applsci-397515). Those comments are all valuable and very helpful for revising and improving our paper. The main corrections in the paper and the response to the reviewer’s comments are as following: 

Reviewer 2 Report

This paper explores the possibility of using GANs to improve the training datasets to obtain a better classification model. To do so, a pipeline based on GANs was implemented by the author of this paper. The authors indicate that based on their experiments and the achieved improvements on the (more robust) AC-GANs superior to prior work results were achieved since the recognition rate can reach more than 95%.

The paper is well-written and the concept well-presented. I also believe that this paper has also a practical applicability in a number of domains that deal with signal classification.

However, there are a few comments that I would like to mention to the authors in order to improve the general presentation and the content of this paper.

My first comment is related to the evaluation of the proposed method. The authors did a good job evaluating heir method 1) in indoor and outdoor environments and 2) using different signals. However, to understand the advantage of the proposed method compared with the state of the art methods, I would recommend authors to expand table 3 by also adding results from different e.g., unsupervised learning techniques such as self-organizing map, deep belief nets, or even more recent and advanced architectures of GANs. Such a comparison would help us understand the superiority of the GANs proposed in this papers.

My second comment is related to the filtered noise. What if the noise if not filtered? Or how does your model behave (compared to other ML techniques) when artificial noise is added?

My third comment is related to the preprocessing of the signals. Generally, it is well known that when pre-processing the extracted features of a signal the accuracy of an ML technique can be increased. So, what is the accuracy of your method if the signal is not pre-processed? How does the pre-processing increase the accuracy? I would also like to ask authors consider pre-processing their signal using RBMs, such as these papers (please consider discussing RBM pre-processing):

-- Feature Preprocessing with RBMs for Music Similarity Learning

-- Learning Motion Features for Example-Based Finger Motion Estimation for Virtual Characters

I feel confident the use of RBM to preprocess your signal might help you improve the accuracy of your method.

Given the amount of work that should be performed by the authors, I am recommending a major revision at this point. I also feel confident that after the authors addressing the comments, the paper will be ready for publication.

Author Response

(The authors gave the same response as above.)

Round 2

Reviewer 2 Report

The revised version of this manuscript addresses all of the raised issue mentioned in the first review cycle. Based on its current form I recommend the paper for publication without further revision.